# The Construction of a Hydrophilic Inorganic Layer Enables Mechanochemically Robust Super Antifouling UHMWPE Composite Membrane Surfaces

**DOI:** 10.3390/polym12030569

**Published:** 2020-03-04

**Authors:** Rong Liu, Shusen Liu, Junrong Yu, Wei Zhang, Jiamu Dai, Yu Zhang, Guangyu Zhang

**Affiliations:** 1School of Textile & Clothing, National & Local Joint Engineering Research Center of Technical Fiber Composites for Safety and Health, Nantong University, Nantong 226019, China; rong063@ntu.edu.cn (R.L.); liuss8089@163.com (S.L.); zhangwei@ntu.edu.cn (W.Z.); jmdai@ntu.edu.cn (J.D.); z.yu@ntu.edu.cn (Y.Z.); zgyu85@ntu.edu.cn (G.Z.); 2State Key Laboratory for Modification of Chemical Fibers and Polymer Materials, College of Materials Science and Engineering, Donghua University, Shanghai 201620, China

**Keywords:** ultra-high-molecular-weight polyethylene, vinyl trimethoxy silane, chemical grafting, nanosilica, antifouling properties

## Abstract

In this study, a facile and effective method is adopted to prepare mechanochemically robust super antifouling membrane surfaces. During the process, vinyl trimethoxy silane (VTMS) was used as the reactive intermediate for coupling the hydrophilic inorganic SiO_2_ nanoparticle layer on to the organic ultra-high-molecular-weight polyethylene (UHMWPE) membrane surface, which created hierarchical nanostructures and lower surface energy simultaneously. The physical and chemical properties of the modified UHMWPE composite membrane surface were investigated. FTIR and XPS showed the successful chemical grafting of VTMS and SiO_2_ immobilization, and this modification could effectively enhance the membrane’s surface hydrophilicity and filtration property with obviously decreased surface contact angle, the pure water flux and bovine serum albumin (BSA) rejection were 805 L·m^−2^·h^−1^ and 93%, respectively. The construction of the hydrophilic nano-SiO_2_ layer on the composite membrane surface for the improvement of membrane antifouling performance was universal, water flux recovery ratio values of BSA, humic acid (HA), and sodium alginate (SA) were all up to 90%. The aim of this paper is to provide an effective approach for the enhancement of membrane antifouling performance by the construction of a hydrophilic inorganic layer on an organic membrane surface.

## 1. Introduction

Currently, with the growing concerns of the worldwide increasing water crisis, numerous researchers are working on the problem. Separation membranes with efficient separation efficiency and excellent antifouling properties have been the key focus technology due to the advantages of reliability, safety, and energy saving [1,2,3,4,5,6]. According to our previous study, an extremely higher water flux, ultra-high-molecular-weight polyethylene (UHMWPE)/fabric composite membrane was succeed fabricated by the composition between UHMWPE and pretreated woven fabric [7]; after the composition, the UHMWPE’s sponge-like non-through pore structure became penetrated, and the porosity and water flux had increased significantly. However, this composite fabrication did not improve the antifouling properties of the UHMWPE composite membrane, due to the UHMWPE’s strong hydrophobic nature and lack of any reactive functional groups. In the meantime, the conditions required to initiate grating by hydrogen abstraction are limited, hence, high-energy radical processes including electron irradiation are used to modified UHMWPE, and those methods are ineffective [8]. Furthermore, there are approaches for obtaining hydrophilic membrane surfaces, such as amine treatment [9], bio-inspiration, grafting [10], solvent adsorption [11], and polymer blending [12]. Therefore, an effective and energy-saving method to enhance the antifouling properties of the UHMWPE composite membrane has become an emergency issue of our next work.

Recently, much attention was focused on the fabrication of a uniformly inorganic hydrophilic layer in improving the antifouling property of a membrane due to the indispensable role the thin layer played during the filtration process [13,14,15,16]. The hydrophilic layer could absorb water and form a protective hydrated sheath to prevent the membrane material surface from contacting and adsorbing pollutants [17,18,19]. For example, Shao et al. [20] reported a facile strategy to enhance polyvinylidene fluoride (PVDF) ultrafiltration membrane performance via self-polymerized polydopamine followed by hydrolysis of ammonium fluotitanate. The test results proved that water flux, bovine serum albumin (BSA) rejection, and antifouling performance of PVDF membranes after surface modifications had been greatly enhanced. Zhao et al. [21] fabricated an extremely ordered and hydrophilic surface layer via self-assembly of anionic polyelectrolyte sodium alginate and amino functionalized SiO_2_ nanoparticles. They found that the surface-modified PVDF membranes exhibited excellent hydrophilicity with obviously decreased surface contact angle, which distinctly enhanced the antifouling properties for humic acids and bovine serum albumin with a secondary water flux recovery ratio (FRR) of nearly 100%. Rahimpour et al. [22] reported a facile strategy to enhance polyethersulfone (PES) ultrafiltration membrane performance via coupling TiO_2_ nanoparticles on the membrane surface through UV irradiation and proved that coating TiO_2_ on the membrane surface is a superior technique for the modification of PES membranes to minimize membrane fouling. It has been shown that the construction of a nanoscale inorganic hydrophilic surface was one of the useful methods to improve membrane antifouling. However, due to the lack of any reactive functional group on the UHMWPE composite membrane surface, the immobilization of inorganic hydrophilic surface on UHMWPE composite membrane is extremely difficult.

So far as we know, organosilanes with multifarious functional groups were originally developed as a method to introduce functionalities to an inorganic substrate [23,24,25,26,27,28]. Organosilane-modified inorganic nanoparticles (include SiO_2_, TiO_2_, carbon nanotube, etc.) were blended into polymeric membranes with the purpose of improving membrane performance [29,30]. For example, Zhou et al. [31] reported a strategy to improve the toughness of melamine-formaldehyde/polyvinyl alcohol (MF/PVA) composite membranes by silane coupling agent KH570-modified SiO_2_ (KH570-SiO_2_), the results shown that KH570-modified SiO_2_ has better dispersion in MF/PVA resin and the tensile properties of KH570-SiO_2_/MF/PVA membrane increased significantly. Park et al. [32] used silane- and epoxy-terminated silicone compounds to modify nanosilica and found that it could be applied to epoxy nanocomposite for outdoor insulating material. Iyer et al. [33] introduced a facile method to achieve highly effective dispersion of agglomerated nanofiller by hexamethyldisiloxane-modified nanosilica. It has been shown that organosilanes could connect the organic polymer and inorganic molecule efficiently.

Therefore, in this paper, the vinyl trimethoxy silane (VTMS) was used to introduced reactive functional groups on the inert UHMWPE composite membrane surface via chemical grafting. Then nanoscale SiO_2_ particles were covalently attach to the organosilane-functionalized membrane surface to fabricate an inorganic hydrophilic surface. According to our understanding, there is rarely strategy reported to construct SiO_2_ nanoparticles uniformly onto the surface of UHMWPE/fabric composite membrane with the organosilanes used as reactive intermediates and tight covalent binding for performance enhancement. The physical and chemical properties of the modified UHMWPE/fabric composite membrane surface were investigated in detail. The pure water flux, pollutant rejection, and antifouling properties to typical different pollutants were also studied.

## 2. Experimental

### 2.1. Materials

UHMWPE with Mη of 4,000,000 was supplied by Beijing Eastern Petrochemical Co., LTD. Liquid paraffin, used as a diluent, was purchased from Hangzhou Refinery, China. Xylene and ethanol, all chemically pure (≥99.5%), were purchased from Sinopharm Chemical Reagent Co., Ltd., China. Irganox1076 (purity 92%), used as the antioxidant, was purchased from Ciba (Switzerland). Hydrophilic gaseous SiO_2_ (15 nm), vinyl trimethoxy silane (VTMS, purity >95%), lauroyl peroxide (LPO, purity 95%), bovine serum albumin (BSA, purity 96%), sodium alginate (SA, purity 95%), and humic acid (HA, fulvic acid >90%) were all purchased from J&K Chemical Technology without further purification (Shanghai, China). Polyester woven fabric was brought from Shanghai Textile Industry, the linear density of the polyester yarn was 90 dtex.

### 2.2. Synthesis of Inorganic Hydrophilic Surface Composite Membrane

UHMWPE/fabric composite membranes were fabricated according to our previous studies [7]. Firstly, a modification solution was prepared by dissolving VTMS and LPO in xylene solution (VTMS 13 wt.%, LPO:VTMS = 1:10), the UHMWPE/fabric composite membranes were immersed into the modification solution and reacted at 75 °C for 4 h to graft VTMS on the membrane surfaces by free radical reaction induced by LPO. Then, the VTMS-grafted composite membranes were obtained and taken out. Thirdly, 3.0 g/L nano-SiO_2_ hydrolysate was prepared by ultrasonic dispersion of nano-SiO_2_ in deionized water/ethanol solution for 30 min (deionized water:ethanol = 1:18), the VTMS-grafted composite membranes were dipped into hydrolysate and hydrolysis reacted at 50 °C for a stipulated binding time. At last, the modified composite membranes were washed with deionized water to thoroughly remove the unreacted nano-SiO_2_ and then carefully stored in deionized water. Figure 1 presents a schematic outlining of the modification procedure. The modification conditions for different membranes are listed in Table 1.

### 2.3. Characterization of Inorganic Hydrophilic Surface Composite Membrane

The microstructure of the grafted membranes was characterized by a field emission scanning electron microscope (FESEM, S-4800, Nitachi, Japan) at 5.0 kV. An attenuated total reflectance Fourier-transform infrared spectroscopy (ATR-FTIR, NEXUS-670, Bruker Optics, Germany) was employed to record the different modified membrane surfaces. The surface chemical compositions of the membranes were analyzed by X-ray photoelectron spectroscopy (XPS, Escalab 250Xi, Thermo Scientific, Waltham, MA, USA) using Al K Alpha as the radiation source, survey spectra of the membranes were collected over a range of 0–1100 eV. The hydrophilicity of the modified composite membrane was measured using a contact angle goniometer (OCA40Micro, Malvern, UK), the droplets volume used for static contact angle measurements was 5 μL.

### 2.4. Membrane Permeation Performance Evaluation

The pure water flux, rejection, and antifouling property were all evaluated by using a cross flow filtration with an effective filtration area of 11.3 cm^2^; all filtration tests were performed at 0.1 MPa at room temperature, the feed pressure was supplied by a water pump [34]. The volume of permeated pure water was collected for a certain time and the stable flux was calculated by Equation (1). The rejection of the membrane was measured using BSA protein and HA as model foulant and was calculated from pollutant concentrations in the feed and the permeate solutions measured via UV-spectrophotometer (UV-9100 D, LabTech, Beijing, China) according to Equation (2).
(1)$$\mathrm{Water flux}=\frac{Q}{A}$$
*Q* is the flux per unit time (L/h), A is the effective area of the membrane (m^2^).
(2)$$\mathrm{Rejection}=\left(1-\frac{c_{p}}{c_{f}}\right)\times 100\%$$
*c_p_* and *c_f_* are the concentrations of pollutant in permeate and feed solutions, respectively.

### 2.5. Antifouling Performance Evaluation

Static adsorption fouling [35] was used to characterize the antifouling properties of the membranes. The membranes were cut into regular shapes and immersed in BSA solution (0.5 g/L, pH 7.4). After the adsorption–desorption reached equilibrium for 12 h at room temperature, the concentrations of BSA solution before and after adsorption were measured via a UV-spectrophotometer (UV-9100 D, LabTech, Beijing, China) and the adsorption mass was calculated using Equation (3).
(3)$$\mathrm{Adsorption mass}=\frac{\left(c_{1}-c_{2}\right)\times V}{A}$$
where *A* is the area of tested membrane, *V* is the volume of BSA solution, *c*_1_ and *c*_2_ are the concentrations of BSA solution before and after adsorption, respectively.

Dynamic filtration tests were performed to further evaluate the antifouling performance of the membranes using BSA (0.5 g/L, pH 7.4), SA (0.5 g/L, pH 7.4), and HA (0.5 g/L, pH 7.4) as model proteins. The four time cycle process was as follows: Firstly, the membrane was kept at 0.1 MPa for 60 min to obtain a stable water flux, the flux was recorded at 0.1 MPa every 5 min and at least 12 readings were collected to obtain an average value as *J*_1_. Secondly, the feed solution was changed to BSA solution (0.5 g/L, pH 7.4), then the first process was repeated, and the average flux was obtained as *J_p_*. Thirdly, the feed solution was changed to pure water, then the first process was repeated again, and the average flux was obtained as *J*_2_. At last, the tested membrane was taken out from the filter system and washed with deionized water for 15 min and the first process was repeated to obtain the tertiary pure water flux as *J*_3_. The flux recovery ratio of unwashed membrane (FR), the flux recovery ratio of washed membrane (FRR), irreversible fouling ratio (IFR), reversible flux decay ratio (RFR), and total flux attenuation ratio (TFR) were calculated to describe the antifouling ability of the membranes using Equations (4)–(8) [36,37,38,39].
(4)$$\mathrm{FR}=\frac{J_{2}}{J_{1}}\;\times \;100\%$$
(5)$$\mathrm{FRR}=\frac{J_{3}}{J_{1}}\;\times \;100\%$$
(6)$$\mathrm{IFR}=\left(\frac{J_{1}-J_{3}}{J_{1}}\right)\times \;100\%\;=\;1-FRR$$
(7)$$\mathrm{RFR}=(\frac{J_{3}-J_{p}}{J_{1}})\;\times \;100\%$$
(8)$$\mathrm{TFR}=\left(\frac{J_{1}-J_{p}}{J_{1}}\right)\;\times \;100\%$$

## 3. Results and Discussion

### 3.1. Characterization of Surface Physicochemical Properties of Modified Composite Membranes

To robustly construct a nano-SiO_2_ hydrophilic layer on the surface of UHMWPE/fabric composite membranes, the original composite membrane had been pre-modified via the chemical grafting of VTMS, which was initiated by LPO. After that, nano-SiO_2_ layer could be uniformly immobilized onto the VTMS-grafted composite membrane by the hydrolysis of silicon on VTMS [40,41,42]. The mechanism of polymerization is illustrated in Figure 1. SiO_2_ with the amount of –OH groups on the membrane surface could increase the membrane’ surface polarity, resulting in a highly hydrophilic membrane surface and absorb water, which could form a protective hydrated sheath to prevent the membrane surface from contacting and adsorbing pollutants.

Therefore, composite M/VTMS/SiO_2_ had a water contact angle (CA) as low as 45° as shown in Figure 2. As we all know, the lower water contact angle illustrates the better hydrophilicity of these membranes. As a contrast, the water contact angle of original UHMWPE/fabric composite membrane was as high as 123°, after VTMS grafting for 4 h, the water contact angle of composite membrane slightly decreased from 123° to 102° due to the hydrolysis of silicon on VTMS, meanwhile, the value of the water contact angle decreased dramatically with the later immobilization of SiO_2_ nanoparticles, and decreased with the increased SiO_2_ binding time. This trend indicates that SiO_2_ with the amount of –OH groups had been successfully bonded onto the composite membrane surface, which was also proved by ATR-FTIR and XPS in Figure 3 and Figure 4.

To demonstrate the modification process and analyze the chemical composition of the modified composite membranes surface, ATR-FTIR was employed to evaluate the original composite membrane, VTMS-grafted composite membrane, and later immobilized SiO_2_. As displayed in Figure 3, the original composite membrane exhibited many strong absorption peaks at 2919, 2851, 1467, and 714 cm^−1^, corresponding to the stretching, bending, and rocking vibrations of –CH_2_ groups. After grafting with VTMS, new peaks of Si–O–Si and Si–O bonds appeared at 1113 and 805 cm^−1^, corresponding to the hydrolysis of silane on VTMS and polycondensation between VTMS, indicating successful grafting reaction. Moreover, the broad absorbance at 3417 cm^−1^ corresponds to stretching vibrations of O–H, which also proved the successful bonding of VTMS and the hydrolysis of silane on VTMS. Lastly, after immobilizing SiO_2_, the peaks of Si–O–Si and Si–O bonds appearing at 1113 and 805 cm^−1^ bands became significantly bigger and sharper, which proved abundant SiO_2_ was successfully immobilized on the composite membrane, and the broader O–H peak at 3417 cm^−1^ also proved that.

Moreover, as depicted in Figure 4, XPS was employed to further analyze the surface chemical compositions of pristine composite membrane and modified membranes. Compared with the pristine composite membrane, new O1s and Si2p peaks appeared on the spectrum of composite M/VTMS, which can be attributed to the oxygen and silicon elements from the graft of VTMS. After immobilization with SiO_2_, the high-resolution O1s peak of the modified membrane exhibited a much stronger peak than the pristine membrane, which indicates that the composite membrane was fully encapsulated by nano-SiO_2_. In addition, the element percentages of C, O, and Si, which are displayed in Table 2, also demonstrate uniform dispersion of SiO_2_ on the membrane surface. After grafting by VTMS, the concentrations of O1s and Si2p on the membrane surface increased to 26.04% and 12.25%, respectively. After immobilization with SiO_2_, the O1s and Si2p concentrations become even higher (35.38% and 24.23%, respectively). This result indicates that SiO_2_ had covered the surface of the investigated membrane.

The narrow scan spectra and curve fitting for C1s, O1s, and Si2p of the neat and modified composite membranes after grafting VTMS and immobilizing SiO_2_ are shown in Figure 5. The C1s of the neat composite membrane surface had only a C–C/C–H peak at 284.7 eV, while the C1s of UHMWPE/fabric/VTMS could be resolved into two peaks at 285.5 and 284.7 eV, respectively, corresponding to C–Si and C–C/C–H. Furthermore, the peaks of composite M/VMTS/nano-SiO_2_ exhibited two similar peaks, the C–C/C–H peak area became smaller, which was because the introduction of nano-SiO_2_ did not bring new C1s peaks, but decreased the C–C/C–H peak area. Meanwhile, the O1s, Si2p peaks of composite M/VMTS and composite M/VMTS/nano-SiO_2_ also showed a similar trend. In Figure 5C,F, we see the peak components at 532.5 and 533.1 eV, which belong to Si–O and O–H, respectively; Figure 5D,G shows there were two peaks at 102.8 and 103.5 eV, respectively, corresponding to Si–C and Si–O, that was also because the introduction of nano-SiO_2_ did not bring new C1s and Si2p peaks. XPS results are consistent with that of the ATR-FTIR spectra. All these substantially prove that VTMS was successfully grafted on the surface of the composite membrane and SiO_2_ was immobilized on the membrane surface via silanol hydrolysis.

The morphological change of membranes before and after modification has been characterized with SEM. Figure 6(M1–M6), shows the typical UHMWPE/fabric composite membrane surface morphology images at 200×, this special structure of composite membrane has been particularly described in our previous studies [7], and there were no obvious distinctions among different membranes at low magnification. However, the enlarged surface views in Figure 6(m1–m6) indicated the existence of modification. Compared with the original composite membrane, the appearance of SiO_2_ nanoparticles could clearly been observed on M3–M6(composite M/VMTS/nano-SiO_2_ with different reacting times: 1.5 h/3 h/6 h/8 h), and with the increased reacting time the amount of SiO_2_ nanoparticles increased, the SiO_2_ layer became much denser and noticeable, the pore size decreased on the membrane surface also. In Figure 6(m5), after 6 h coating, the big pores on the membrane surface were distinctly diminished, which indicated that coupling inorganic SiO_2_ nanoparticles layers on composite membrane surface could effectively reduce the surface porosity of the membrane, which will conspicuously affect the membrane separation performance.

### 3.2. Separation Performance

SEM/water contact angle/FTIR and XPS results proved that a hydrophilic and denser inorganic nanoparticle layer was successfully fixed onto the composite membrane surface, which will conspicuously affect the membrane separation properties. The membrane separation properties were evaluated by pure water flux and BSA rejection. As shown in Figure 7A,B, the pure water flux was 452.2 L·m^−2^·h^−1^ and rejection for HA/BSA was 36% and 78%, respectively, for original composite membrane. After grafting with VTMS, the pure water flux of M2 was increased to 532.7 L·m^−2^·h^−1^, and the rejection of HA/BSA was increased to 42.3% and 82%, respectively, which was due to a small quantity of –OH brought by the hydrolysis of silicon on VTMS. Meanwhile, the water flux of M3, M4, M5, and M6 membranes were dramatically increased to 621, 756.6, 805, and 815 L·m^−2^·h^−1^, respectively, and the rejection of HA/BSA was increased to 51%/81%, 62%/89%, 71%/91%, and 73%/93%, respectively. The results revealed that modified composite membranes had evident separation properties, which indicated that the immobilization of hydrophilicity and denser SiO_2_ nanoparticles layer on the membrane surface played an important role in optimizing separation performance. In general, after fixing a hydrophilic inorganic nanoparticle layer on the membrane surface, the membrane’s surface properties had been changed from hydrophobic to hydrophilic; therefore, during the separation process, water molecules could be spontaneously adsorbed on the membrane surface, which would act as an activity process to promote the pure water flux [43]. Meanwhile, the denser and ordered SiO_2_ nanoparticle layers have efficiently minimized the membrane surface pore size, which could increase the rejection of BSA and HA molecules.

The SiO_2_ nanoparticle-grafted composite membranes obviously exhibited positive effects on the variation of water flow during the preloading process, as depicted in Figure 8A. The stabilization flux of neat composite membrane and composite M/VMTS could be obtained after 40 min of preloading; however, after immobilization with SiO_2_ nanoparticles, the time to obtain a stable flux was conspicuously shortened to 20 min, which was attributed to the immobilization of the impact hard inorganic SiO_2_ layer. As we all know, during the separation process, loosened membrane pores would be compacted under the applied operation pressure, compared with polymer long chain, those SiO_2_ nanoparticles could rapidly rearranged to steady states under extra pressure, leading to shorter preloading time. Meanwhile, the bonding fastness of nanoparticles on the membrane surface was also evaluated, as shown in Figure 8B. After 40 h of continuous cross flow filtration, the pure water flux of composite M/VMTS/nano-SiO_2_/6 h remained stable, which proved that the bond between the nanoparticles and the membrane surface was a strong chemical bond rather than physical adsorption.

### 3.3. Antifouling Performance

The effect of the hydrophilic inorganic SiO_2_ nanoparticle layer for antifouling properties was investigated in detail in this part. In general, the evaluation of antifouling performance includes pollutant static adsorption fouling and dynamic fouling tests. The static adsorption fouling test for neat and modified membranes used BSA as the pollutant and is presented in Figure 9. The Static adsorption mass of neat UHMWPE/fabric composite membrane was 50.4 μg/cm^2^; after grafting with VTMS, the static adsorption mass slightly decreased to 42 μg/cm^2^, due to the small amount of –OH brought by hydrolysis of VTMS. However, silanol on VTMS was very unstable and prone to intermolecular crosslinking, therefore, although the graft of VTMS could improve the hydrophilicity of the composite membrane to some extent, the improvement was not significant. Subsequently, SiO_2_ nanoparticles were covalently immobilized onto the surface of composite membrane, the BSA adsorption capacity of modified composite membrane (M3–M6) dramatically decreased to 34, 15, 3.4, and 3.2 μg/cm^2^, respectively. It could be clearly seen that after 6 h reacting, the composite membrane exhibited excellent anti-protein adsorption abilities. This was because it took time for the silane of VTMS hydrolyze into silanol and bond with SiO_2_ nanoparticles, as the reaction time reached 6 h, adequate nano-SiO_2_ particles were completely assembled on the surface of the composite membrane. Moreover, the presence of the hydrophilic SiO_2_ inorganic nanoparticle layer obviously inhibited the non-specific adsorption of pollutant protein molecules. Thus, the antifouling properties of the membranes grew with the growth of nano-SiO_2_ on composite membrane surface, and it reached the highest value when the nano-SiO_2_ immobilizing time reached 6 h.

A dynamic fouling test was used to further evaluate the hydrophilic inorganic SiO_2_ layer’s antifouling performance. Figure 10 presents the membrane’s time-dependent flux variation during four cyclic dynamic filtration processes. During the first cyclic test, which used pure water as the feeding solution, compared with the neat composite membrane, the pure water flux of VTMS-grafted membranes increased, meanwhile, the pure water flux of composite M/VTMS/SiO_2_ increased significantly. After change the feeding solution to BSA pollutant, the flux rapidly decreased due to the increased solution concentration and protein fouling, but the flux of SiO_2_-modified composite membrane was still kept at its highest. The third and fourth cycles and Figure 10B were used to evaluated non-washed (FR)/washed(FRR) flux recovery value of neat composite membrane and modified membrane after filtration with BSA pollutant, it could be clearly found that, after membrane washing, the non-washed FR/washed FRR value of neat composite membrane increased from 34% to 48%, the irreversible pollution was 52%. After grafting with VTMS, the FR and FRR value only slightly increased from 48% to 57%. However, the FR and FRR of SiO_2_-nanoparticle-immobilized composite membrane were all remarkably high, 71% and 90%, respectively, the irreversible pollution was only 10.4%. Therefore, the modified membrane with hydrophilic and denser nano-SiO_2_ layer could significantly reduce the irreversible pollution and restrain the absorption of protein pollutants.

Table 3 summarizes some of the antifouling membranes reported which involve different modification methods. It has been shown that the construction of nanoscale inorganic hydrophilic surface was one of the useful methods to improve the membrane antifouling. Compared with recent reports, our work was facile and effective. Many studies [44,45,46] had proved that the construction of uniform density and hydrophilic nanoparticle coatings on membrane surface could exhibit superior antifouling property. This hydrophilic layer could absorb water and form a protective hydrated sheath to prevent the membrane material surface contacting and adsorbing pollutants [19,21]; hence, pollutant adsorption and accumulation on membrane pores was greatly reduced leading to extremely less irreversible membrane fouling. On other hand, the dynamic fouling properties were measured by cross flow filtration. During the filtration process, the feed solution parallel to the membrane surface, and the hydrophilic inorganic particle layer on the membrane surface formed an interfacial hydration layer based on hydrogen bonding interaction with water molecules, to block the membrane surface’s direct contact with pollutants. Therefore, the shear force generated by the liquid flowing through the nano-SiO_2_-modified membrane surface could further remove the trapped contaminants. The schematic of antifouling effect ability of nano-SiO_2_ constructed hydrophilic membrane surface as shown in Figure 11. Hence, non-washed composite M/VMTS/nano-SiO_2_/6 h membrane also exhibited excellent antifouling property, and after washing, the proportion of the membrane flux decrease dramatically reduced.

In the process of sewage treatment, the membrane will directly contact and separate various pollutant systems, and the antifouling ability during filtration is the real manifestation of membrane antifouling performance. In the polluted water system, there are some other natural pollutants such as polysaccharide pollutants and humic acid beside protein contaminants. Therefore, two typical pollutants sodium alginate (SA) and humic acid (HA) were used to represented polysaccharide contaminants and natural pollutants in sewage for antifouling tests [47]. The results of the dynamic filtration test of the membrane before and after modification, which used HA/SA as pollutants, are shown in Figure 12. The flux recovery ratio of washed membrane (FRR), irreversible fouling ratio (IFR), reversible flux decay ratio (RFR), and total flux attenuation ratio (TFR) were used to characterize the membrane’s antifouling ability. In general, the higher the FRR and RFR values, or the lower the IFR and TFR, the better the membrane’s antifouling performance. As depicted in Figure 12A, The IFR value of the composite membrane/VTMS/nano-SiO_2_/6 h membrane was only 2%, the irreversible pollution was greatly reduced, and exhibited the highest RFR value. Meanwhile, as shown in Figure 12B, the FRR and IFR values of the SA pollution system presented the same trend as the HA system. The hydrophobic original composite membrane and VTMS-grafted composite membrane were more prone to membrane fouling, while the composite membrane/VTMS/nano-SiO_2_/6 h with significantly improved hydrophilicity showed higher FRR values for both pollutants, no less than 95%, which indicated that the construction of the hydrophilic nano-SiO_2_ layer on composite membrane for the improvement of the antifouling performance was universal.

In general, the significant improvement of composite membrane antifouling performance was attributed to the construction of the hydrophilic nano-SiO_2_ layer on the membrane surface. Its excellent hydrophilicity and hydration ability could achieve comprehensive protection of the composite membrane surface and membrane pores. The nano-SiO_2_ hydrophilic layer was formed by the interaction of hydrogen bonding and water molecules to form an interfacial hydration layer, which not only prevented the contaminants from approaching the composite membrane surface, but also reduced the acting site between contaminant and membrane surface, the weakened anchoring action, and could not support large-scale contaminant aggregation. Therefore, even if a small amount of pollutants was adsorbed on the composite membrane surface, it could be washed away under the shearing action of the water flow during the cleaning process.

## 4. Conclusions

In this study, a stable, efficient, inorganic antifouling membrane surface was constructed via chemically grafting inorganic SiO_2_ nanoparticles layer on to the organic UHMWPE membrane surface, VTMS was used as the reactive intermediate for coupling nano-SiO_2_, and LPO was used as the redox initiator. The enhanced nanoscaled coarse structures combined with high surface energy endowed the neat hydrophobic UHMWPE composite membrane with excellent filtration properties and terrific hydrophilicity with a water contact angle lower than 45°. The pure water flux and BSA rejection of the nano-SiO_2_-modified composite membrane were 805 L·m^−2^·h^−1^ and 93%, the values are steady-state. Furthermore, the modified UHMWPE composite membrane exhibited remarkable and universal antifouling performance, FRR values were 90%, 98%, and 94% for BSA, HA, and SA as typical pollutants, respectively. This article provided a significant improvement of the antifouling performance of hydrophobic UHMWPE composite membrane which can be attributed to the construction of the hydrophilic nanostructure on the membrane surface. This facile and effective technique can be adopted by industry for UHMWPE and other inert material finishing and employed on various substrates for energy applications with customized configurations.

## Figures and Tables

**Figure 1 polymers-12-00569-f001:**
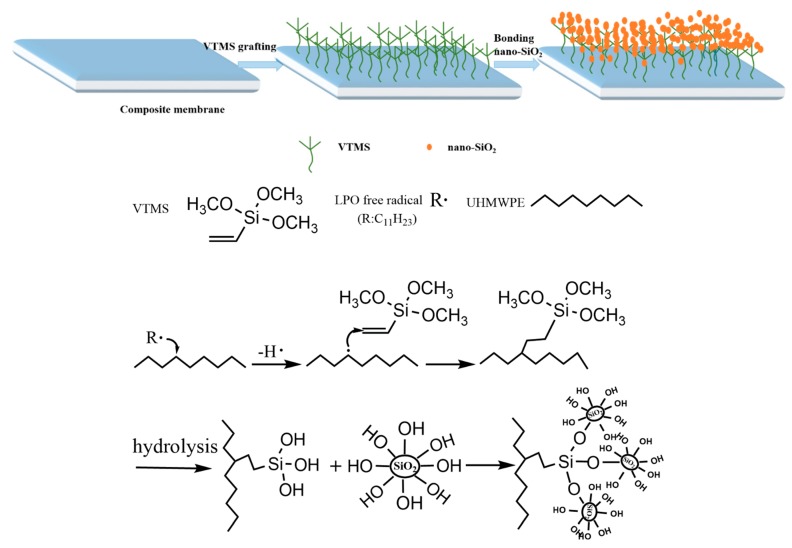
The mechanism for nano-SiO_2_ modified ultra-high-molecular-weight polyethylene (UHMWPE)/fabric composite membrane.

**Figure 2 polymers-12-00569-f002:**
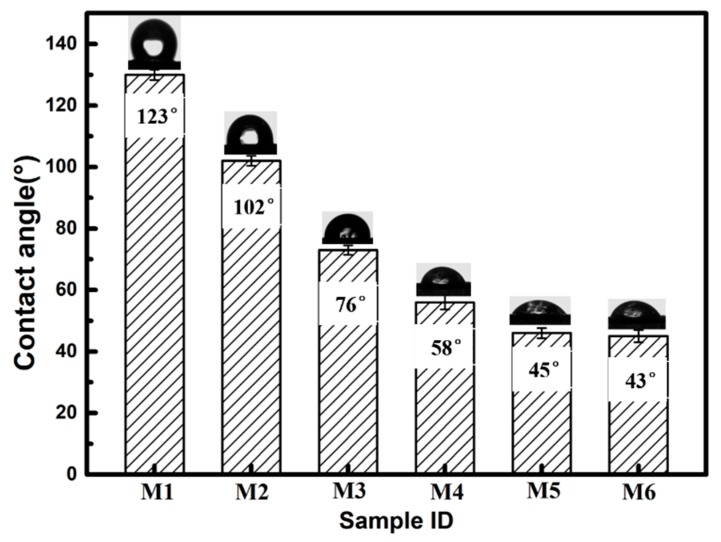
Water contact angle images of various membranes (M1: original UHMWPE/fabric composite membrane, M2: composite M/VMTS, and M3–M6: composite M/VMTS/nano-SiO_2_ (different reacting times: 1.5 h/3 h/6 h/8 h).

**Figure 3 polymers-12-00569-f003:**
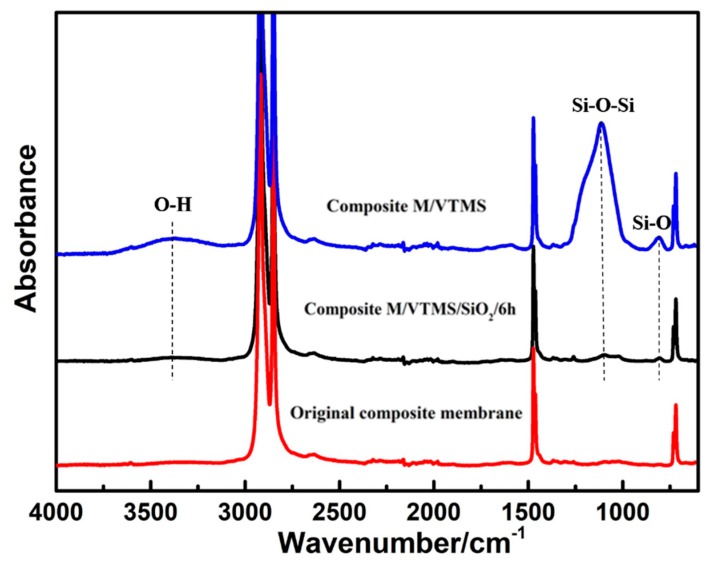
Attenuated total reflectance Fourier-transform infrared spectroscopy (ATR-FTIR) spectrums of the original UHMPWE/fabric composite membrane and surface-modified membranes with VTMS and nano-SiO_2_.

**Figure 4 polymers-12-00569-f004:**
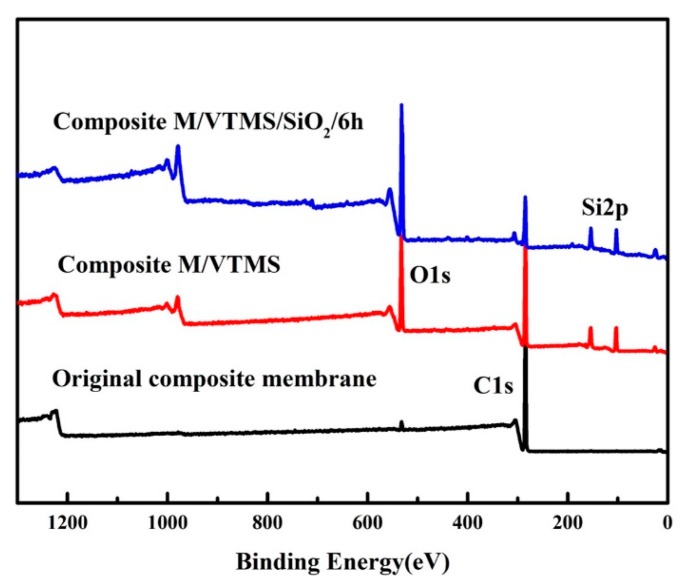
XPS spectra of the original UHMPWE/fabric composite membrane and modified membranes with VTMS and nano-SiO_2_.

**Figure 5 polymers-12-00569-f005:**
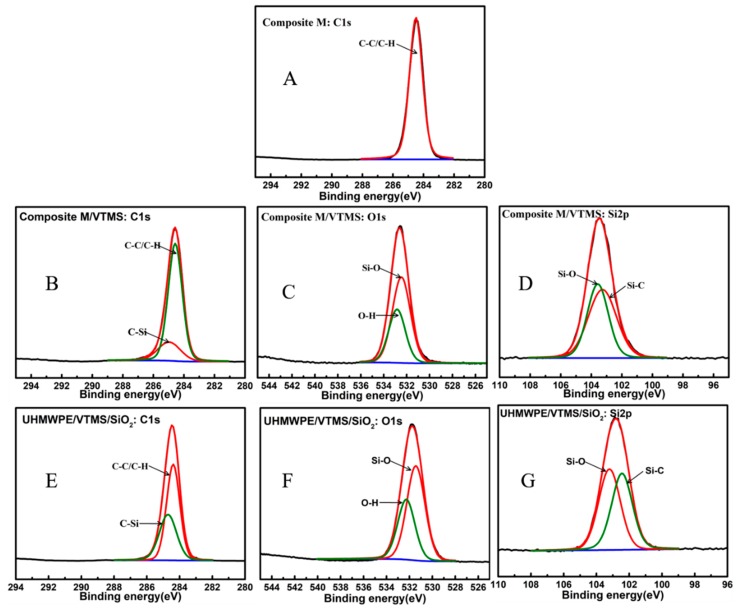
XPS narrow scan spectra of the original UHMWPE/fabric composite membrane and modified membranes after grafting VTMS and immobilizing nano-SiO_2_: C1s (**A**,**B**,**E**), O1s (**C**,**F**), and Si2p (**D**,**G**).

**Figure 6 polymers-12-00569-f006:**
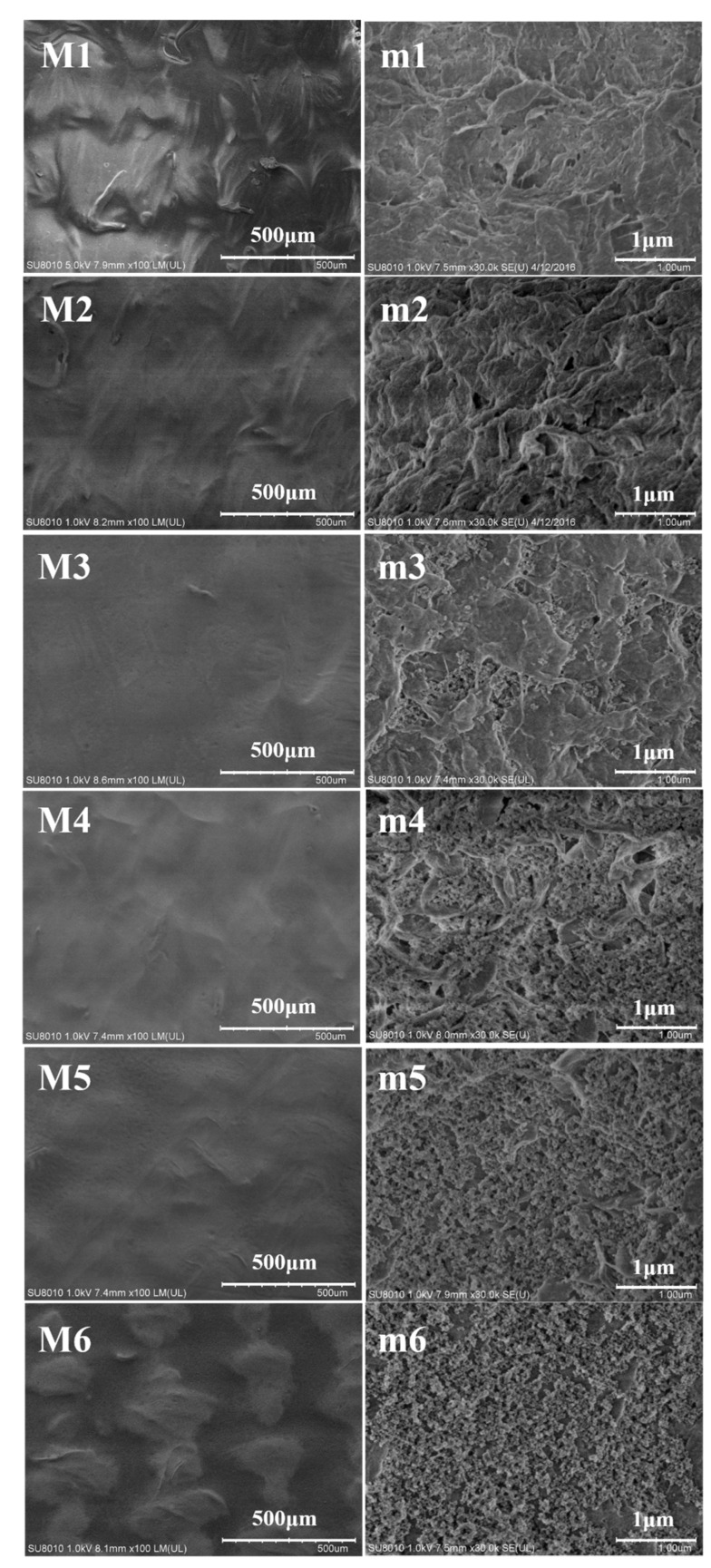
SEM surface morphology images of membranes before and after modification (M1/m1: original UHMWPE/fabric composite membrane, M2/m2: composite M/VMTS, and M3/m3–M6/m6: composite M/VMTS/nano-SiO_2_ (different reacting times: 1.5 h/3 h/6 h/8 h).

**Figure 7 polymers-12-00569-f007:**
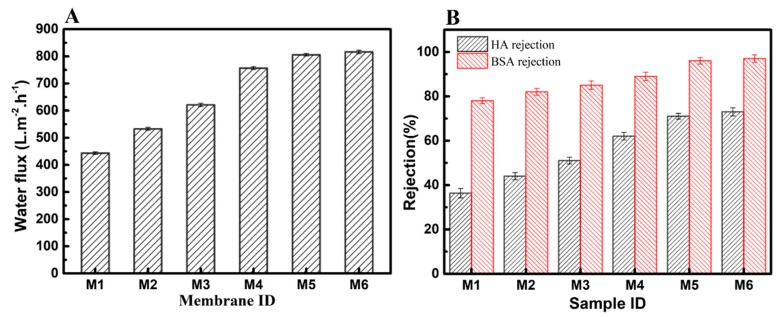
Water flux (**A**) and rejection (**B**) of various membranes (M1: original UHMWPE/fabric composite membrane, M2: composite M/VMTS, and M3–M6: composite M/VMTS/nano-SiO_2_ (different reacting times: 1.5 h/3 h/6 h/8 h).

**Figure 8 polymers-12-00569-f008:**
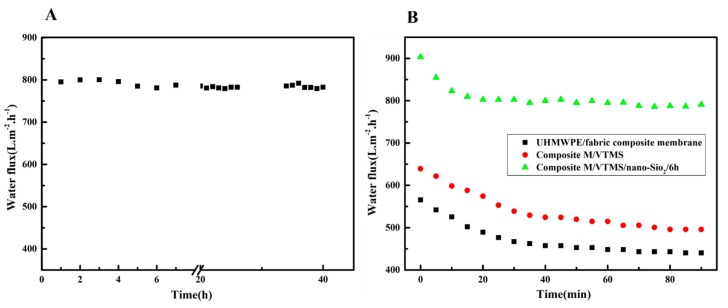
(**A**) Time-dependent separation performance of original UHMWPE/fabric composite membrane, composite M/VMTS, and composite M/VMTS/nano-SiO_2_/6 h, (**B**) long-term separation test of composite M/VMTS/nano-SiO_2_/6 h.

**Figure 9 polymers-12-00569-f009:**
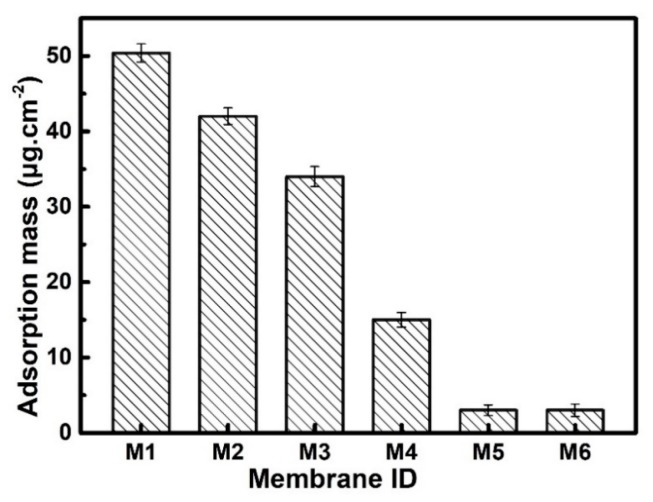
Static adsorption mass of bovine serum albumin (BSA) for neat and modified composite membranes (M1: original UHMWPE/fabric composite membrane, M2: composite M/VMTS, and M3–M6: composite M/VMTS/nano-SiO_2_) different reacting times: 1.5 h/3 h/6 h/8 h.

**Figure 10 polymers-12-00569-f010:**
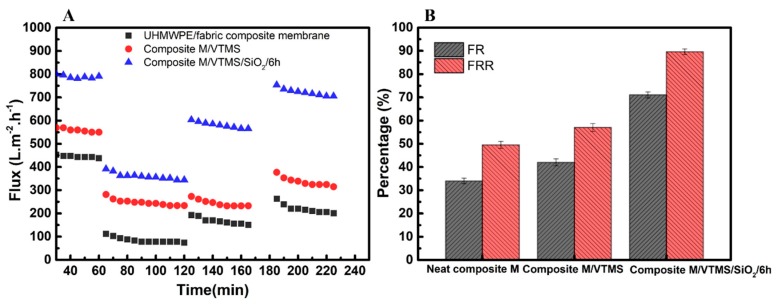
Antifouling performances of all the membranes with BSA as pollutant (**A**): Time-dependent flux variation, (**B**): flux recovery ratio of unwashed membrane (FR) and flux recovery ratio of washed membrane (FRR) values.

**Figure 11 polymers-12-00569-f011:**
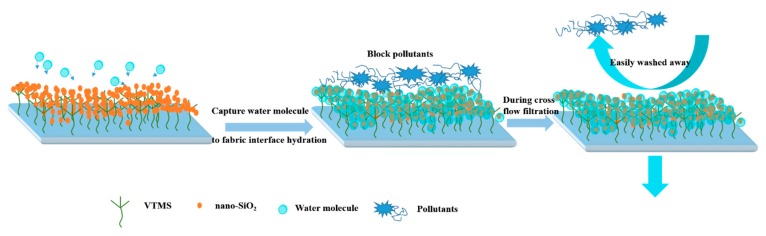
Schematic of antifouling effect ability of nano-SiO_2_ constructed hydrophilic membrane surface.

**Figure 12 polymers-12-00569-f012:**
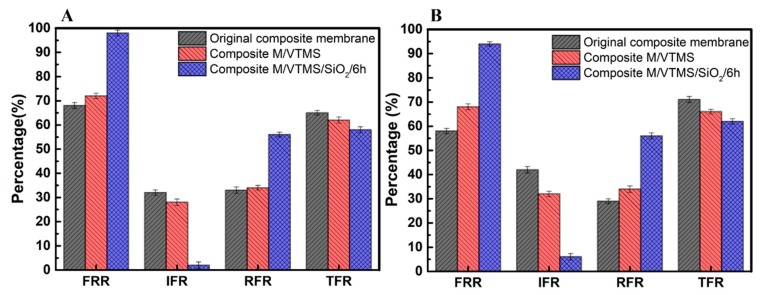
Antifouling property of membranes to humic acid (HA) (**A**) and sodium alginate (SA) (**B**).

**Table 1 polymers-12-00569-t001:** The modification conditions of UHMWPE/fabric composite membranes.

Code	Membrane	Coating Time	
		Step 1 Grafting VTMS (h)	Step 2 nano-SiO_2_ (h)
M1	Composite M	0	0
M2	Composite M/VTMS	4	0
M3	Composite M/VTMS/SiO_2_/1.5	4	1.5
M4	Composite M/VTMS/SiO_2_/3	4	3
M5	Composite M/VTMS/SiO_2_/6	4	6
M6	Composite M/VTMS/SiO_2_/8	4	8

**Table 2 polymers-12-00569-t002:** Surface elemental percentages of the original UHMPWE/fabric composite membrane and modified membranes after grafting VTMS and immobilizing nano-SiO_2_ (data are the averages of three measurements, ±0.1).

Sample	C1s (%)	O1s (%)	Si2p (%)
Original composite membrane	97.71	1.91	0.4
Composite M/VTMS	61.09	26.04	12.25
Composite M/VTMS/SiO_2_/6	40.39	35.38	24.23

**Table 3 polymers-12-00569-t003:** Comparison of modification method, contact angle and BSA flux recovery ratio (FRR) of different membranes for some previously reported similar research.

Membrane	Modification Method	Contact Angle	BSA FRR	Ref
PVDF	Self-polymerized polydopamine and subsequent hydrolysis of TiO_2_	48°	90%	[20]
PVDF	Plasma treatment	33.2°	99%	[21]
PES ultrafiltration membranes	Coupling TiO_2_ nanoparticles with UV irradiation	53.5°	79%	[22]
UHMWPE/fabric composite M	Chemical grafting	43°	93%	Author’s

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
