# Peer review of "The Construction of a Hydrophilic Inorganic Layer Enables Mechanochemically Robust Super Antifouling UHMWPE Composite Membrane Surfaces"

_polymers, 2020, doi:10.3390/polym12030569_

Round 1
Reviewer 1 Report
The manuscript by Liu and co-workers details the preparation, surface modification of nanocomposite membranes with antifouling properties. The topic is timely and of interest to a broad audience with various backgrounds. The subject of the manuscript fits the scope of the journal well. However, there are several minor and major issues to be addressed prior to making a final decision.
1, The first time UHMWPE is used, it should be spelled out (both in the abstract and in the main text).
2, The purity and/or grade for all chemicals, solvents and materials used in the study should be given under the materials section of the manuscript.
3, The main drawbacks and limitations of the proposed methodology should be discussed in the manuscript. In this context, the general applicability of the methodology should be presented. Demonstrate how the methodology can be applied to other scenarios and that it is of interest to a broad audience. This will help to increase the potential impact of the publication.
4, The authors should briefly mention the wide approaches for obtaining hydrophilic membrane surfaces, as a general introduction, with broad examples: amine treatment (10.1016/j.seppur.2017.05.022), bio-inspiration (10.1021/acsnano.8b04123), grafting (10.1016/j.seppur.2018.11.036), solvent adsorption (10.1021/acsami.7b01879), polymer blending (10.1021/acsanm.8b01563), plasma treatment (10.1016/j.memsci.2019.117225).
5, Figure 1 should incorporate the chemical structures and reactions. This will help the readers to understand at molecular level what is shown on the schematic illustration.
6, Some justification should be provided for the selection of the coating times. Why was the 2nd step varied significantly, while the 1st step was not changed?
7, If possible, errors and standard deviations should be provided for the measurements. For instance, Table 2 has no errors and therefore it is not clear whether the observed changes are significant or they are within the error of the measurements.
8, There is closely related previous literature on VTMS for membrane surfaces, these should be briefly acknowledged (10.1016/j.seppur.2010.08.017 for PDMS; 10.1016/j.mspro.2015.04.092 for mixed matrix membranes; 10.1016/j.memsci.2019.117708 for membrane distillation).
9, In Figures 8a and 9, instead of flux, give permeance on the y axis. That is more informative then the flux. Make a comparison with the state-of-the-art literature to demonstrate what improvement has been achieved. In panel b) disclose the molecular weight of the solutes in the figure caption to facilitate understanding the work.
10, In line 384, clarify if the reported values are steady-state or not. Report permeance instead of flux.
Author Response
1, The first time UHMWPE is used, it should be spelled out (both in the abstract and in the main text).
Response: It has been amended and thank you for your advice
2, The purity and/or grade for all chemicals, solvents and materials used in the study should be given under the materials section of the manuscript.
Response: Thank you very much. Related purity/or grade have been added
3, The main drawbacks and limitations of the proposed methodology should be discussed in the manuscript. In this context, the general applicability of the methodology should be presented. Demonstrate how the methodology can be applied to other scenarios and that it is of interest to a broad audience. This will help to increase the potential impact of the publication.
Response: Thank you very much. Related drawbacks and limitations have been added
4, The authors should briefly mention the wide approaches for obtaining hydrophilic membrane surfaces, as a general introduction, with broad examples: amine treatment (10.1016/j.seppur.2017.05.022), bio-inspiration (10.1021/acsnano.8b04123), grafting (10.1016/j.seppur.2018.11.036), solvent adsorption (10.1021/acsami.7b01879), polymer blending (10.1021/acsanm.8b01563), plasma treatment (10.1016/j.memsci.2019.117225).
Response: It has been added and thank you for your advice
5, Figure 1 should incorporate the chemical structures and reactions. This will help the readers to understand at molecular level what is shown on the schematic illustration.
Response: Thank you for your advice, and it has been amended, we combined fig 1 and fig 2
6, Some justification should be provided for the selection of the coating times. Why was the 2nd step varied significantly, while the 1st step was not changed?
Response: 1st step was chemical grafting of VTMS, VTMS was small molecule and the chemical grafting of VTMS didn’t change the membrane surface morphology, but XPS and FTIR all clearly proved the existence of VTMS after 1st step.
7, If possible, errors and standard deviations should be provided for the measurements. For instance, Table 2 has no errors and therefore it is not clear whether the observed changes are significant or they are within the error of the measurements.
Response: we clarified that the data are the averages of three measurements, and error ±0.1
8, There is closely related previous literature on VTMS for membrane surfaces, these should be briefly acknowledged (10.1016/j.seppur.2010.08.017 for PDMS; [1] for mixed matrix membranes; 10.1016/j.memsci.2019.117708 for membrane distillation).
Response: Thank you for your advice and it has been acknowledged
9, In Figures 8a and 9, instead of flux, give permeance on the y axis. That is more informative then the flux. Make a comparison with the state-of-the-art literature to demonstrate what improvement has been achieved. In panel b) disclose the molecular weight of the solutes in the figure caption to facilitate understanding the work.
Response: It has been amended and thank you for your advice
10, In line 384, clarify if the reported values are steady-state or not. Report permeance instead of flux.
Response: It has been amended and thank you for your advice
Reviewer 2 Report
Manuscript: The construction of hydrophilic inorganic layer enables mechanochemically robust superantifouling UHMWPE composite membrane surfaces
Manuscript ID: polymers-731800
Manuscript presents very good research work related to surface modification of composite membrane and can be accepted for publication after minor revision.
- Author need to incorporate some interesting structural characterization data in the abstract part of the manuscript.
- Authors need to incorporate some crescent references related to the subject in introduction part; specially related to water purification technologies. For example;
- Biomacromolecules 18 (8), 2333-2342 (b) ACS Sustainable Chemistry & Engineering 6 (3), 3279-3290 (c) Industrial & Engineering Chemistry Research 56 (46), 13885-13893 (d) ACS Sustainable Chem. Eng., 2019, 7 (6), pp 6140–6151 (e) ACS Omega 2019, 4, 26, 22008-22020 (f) RSC Advances 9 (69), 40565-40576
- Authors need to include exact contact angle values in figure 3 for each sample.
- Authors need to label the important peaks in figure 4 along with functional groups.
- It will be good if authors can include some thermal characterization of membrane samples (TGA).
- Is there any change in surface charge after surface medication of the membranes?
- Author need to compare their results with some of the previously reported simila research in tabulated form.
- Author need to add future prospective of presented research in the conclusion part of the manuscript.
Author Response
Manuscript presents very good research work related to surface modification of composite membrane and can be accepted for publication after minor revision.
- Author need to incorporate some interesting structural characterization data in the abstract part of the manuscript.
- Response: It has been added and thank you for your advice
- Authors need to incorporate some crescent references related to the subject in introduction part; specially related to water purification technologies. For example;
Biomacromolecules 18 (8), 2333-2342 (b) ACS Sustainable Chemistry & Engineering 6 (3), 3279-3290 (c) Industrial & Engineering Chemistry Research 56 (46), 13885-13893 (d) ACS Sustainable Chem. Eng., 2019, 7 (6), pp 6140–6151 (e) ACS Omega 2019, 4, 26, 22008-22020 (f) RSC Advances 9 (69), 40565-40576
- Response: It has been added and thank you for your advice
- Authors need to include exact contact angle values in figure 3 for each sample.
- Response: It has been added and thank you for your advice
- Authors need to label the important peaks in figure 4 along with functional groups.
- Response: Thank you for your advice and it has been amended
- It will be good if authors can include some thermal characterization of membrane samples (TGA).
- Response: TGA data of the composite membrane have been given in my previous work (Rsc Adv, 6 (2016) 90701-90710), and the immobilization of SiO2 have not evidently change the membrane thermal properties.
- Is there any change in surface charge after surface medication of the membranes?
- Response: There was no significant change in charge on the membrane surface.
- Author need to compare their results with some of the previously reported similar research in tabulated form.
- Response: Thank you for your advice and it has been amended
- Author need to add future prospective of presented research in the conclusion part of the manuscript.
- Response: It has been added and thank you for your advice
Round 2
Reviewer 1 Report
1, Previous comment #9. The authors simply changed the title from Flux to Permeance. This is erroneous and must be corrected. Flux is given as L m-2 h-1, while permeance is L m-2 h-1 bar-1. The authors need to divide all flux values with the applied pressure (which currently not given in the figure). Also report the applied pressure in the figure caption. Such error in an article results in serious misinterpretation of the results.
2, The reference list has numerous errors and typos, missing names, publication year, pages, volumes, author names etc. Examples include reference 3 (vol, pages missing), 9 (year missing), 10 (names are incomplete and not abbreviated, journal name, vol, year, pages all missing), 12 (year missing), 13 (vol, year, pages are missing). Moreover, instead of random names, follow the CAS journal abbreviations (https://cassi.cas.org/search.jsp).
Author Response
Dear Editor:
Thank you very much for your detailed comments and suggestions for our manuscript .
I have revised the paper with the suggestions the reviewers offered one by one, and answered the questions one by one below.
If there is any question, please don't hesitate to let me know
Yours sincerely,
Rong Liu
Point to point response
1, Previous comment #9. The authors simply changed the title from Flux to Permeance. This is erroneous and must be corrected. Flux is given as L m-2 h-1, while permeance is L m-2 h-1 bar-1. The authors need to divide all flux values with the applied pressure (which currently not given in the figure). Also report the applied pressure in the figure caption. Such error in an article results in serious misinterpretation of the results.
Response: Thank you so much for your advice, We described the applied pressure in line 125," all filtration tests were performed at 0.1 MPa at room temperature". Meanwhile, numerous of articals use the expression of "flux" to evalute the membrane filtration performance,e.g. ref.10.16.17.18.19.20.21.22.23 and so on, The expression of flux is widely used in membrane filtration articles. so we decide not change the expression of flux, hope your understanding.
2, The reference list has numerous errors and typos, missing names, publication year, pages, volumes, author names etc. Examples include reference 3 (vol, pages missing), 9 (year missing), 10 (names are incomplete and not abbreviated, journal name, vol, year, pages all missing), 12 (year missing), 13 (vol, year, pages are missing). Moreover, instead of random names, follow the CAS journal abbreviations (https://cassi.cas.org/search.jsp).
Response: It has been amended and thank you for your advice
